# Extracting Flooded Roads by Fusing GPS Trajectories and Road Network

**Shiying She [1], Haoyu Zhong [2],\* , Zhixiang Fang [2] , Meng Zheng [3] and Yan Zhou [4]**

[1]   College of Metropolitan Transportation, Beijing University of Technology, Beijing 200124, China; shesy@bjtrc.org.cn
[2]   State Key Laboratory of Information Engineering in Surveying, Mapping and Remote Sensing, Wuhan University, 129 Luoyu Road, Wuhan 430079, China; zxfang@whu.edu.cn
[3]   Wuhan Transportation Development Strategy Institute, Wuhan 430017, China; zhengmeng@whtpi.com
[4]   Department of Urban and Rural Planning, School of Urban Design, WuHan University, 129 Luoyu Road, Wuhan 430079, China; joyeezhou@whu.edu.cn
\*   Correspondence: hyzhong@whu.edu.cn

**Abstract:** Urban roads are the lifeline of urban transportation and satisfy the commuting and travel needs of citizens. Following the acceleration of urbanization and the frequent extreme weather in recent years, urban waterlogging is occurring more than usual in summer and has negative effects on the urban traffic networks. Extracting flooded roads is a critical procedure for improving the resistance ability of roads after urban waterlogging occurs. This  paper proposes a flooded road extraction method to extract the flooding degree and the time at which roads become flooded in large urban areas by using global positioning system (GPS) trajectory points with driving status information and the high position accuracy of vector road data with semantic information. This method uses partition statistics to create density grids (grid layer) and uses map matching to construct a time-series of GPS trajectory point density for each road (vector layer). Finally, the fusion of grids and vector layers obtains a more accurate result. The experiment uses a dataset of GPS trajectory points and vector road data in the Wuchang district, which proves that the extraction result has a high similarity with respect to the flooded roads reported in the news. Additionally, extracted flooded roads that were not reported in the news were also found. Compared with the traditional methods for extracting flooded roads and areas, such as rainfall simulation and SAR image-based classification in urban areas, the proposed method discovers hidden flooding information from geospatial big data, uploaded at no cost by urban taxis and remaining stable for a long period of time.

**Keywords:** flooded road extraction; GPS trajectory points; multi-source data fusion; vector road network

## 1. Introduction

Roads are the backbone of urban transportation. Smooth traffic is crucial for daily commute and emergency travel of the urban residents [1]. Traditional city design and construction often ignore the importance of urban surface permeability, so vegetation areas with good water storage and drainage are covered by a large amount of urban construction materials, which cause the urban water drainage ability to decline sharply [2]. With the continuous increase of extreme climates, some cities have serious waterlogging problems during periods of abnormal precipitation in the spring and summer [3,4]. Roads in low-lying areas or with poor drainage systems are easily flooded, which cause an increase in urban traffic pressure and ultimately incur negative effects on the travel of urban residents. It is of great importance to study the extraction of flooded roads.

To improve the resistibility of urban roads during abnormal precipitation, flooded roads in urban areas need to be detected. Therefore, how flooded roads in a large urban area can be accurately detected has been a serious problem for a long time [5]. In recent years, a common strategy used to extract flooded roads has been to simulate urban precipitation and establish urban rainfall distribution models [6]. Based on the model, vector road data are overlaid on a rainfall distribution grid to determine which roads are located in heavily inundated areas. Although digital elevation model (DEM) and urban drainage systems are considered in the models to improve the precision of extraction results, it is still very hard to simulate the actual influence of urban sewerage systems on flooding distribution because of the different drainage capacities of different places. Therefore, many hypotheses must be considered before simulations start and the results of such simulations are idealized. In addition, radar remote sensing (SAR) images have become a popular data resource to extract flooded areas at a large scale because of their strong ability to avoid cloud disturbances and because of their high resolution, so another way to solve this problem is to classify two-day radar remote sensing images and extract different areas by change detection [7–9]. However, the problem with such data sources is that it is difficult to obtain it in time and that it is hard to measure the flooding degree and time.

As urban taxis begin to be equipped with GPS devices, GPS devices record and upload the position and driving status of taxis in service at intervals [10]. Every day, tens of thousands of urban taxis in service upload millions of GPS track points which directly reflect the actual situation of urban roads, and have played a significant role in urban road network extraction [11,12], vehicle behavior patterns analysis [13–15], road status detection [16], and so on. However, few studies focus on extracting the flooded roads by using GPS trajectory data. Because GPS trajectory points contain semantic information such as driving speed and direction, it is easy to find abnormal conditions such as low traffic flow and low speed on urban roads through the analysis of datasets. To extract flooded roads, this paper proposes a method that takes advantage of multi-source data. This method adopts a three-step extraction and verification strategy by comparing the differences of GPS trajectory points between sunny days and rainy days. The overall strategy can be divided into three steps: calculating the probable inundated area in a grid, calculating the probable flooded roads based on GPS points and vector road data, and calculating and verifying the flooding degree and time based on the fusion of grid and vector data. In contrast to traditional methods, the proposed method makes full use of multi-source data and avoids various assumptions the simulation needs, which leads to a more direct result with higher accuracy and efficiency.

This paper is structured as follows. The literature related to flooded road extraction is discussed in Section 2. Section 3 describes the study area and data in this paper, and the proposed method is introduced in detail in Section 4. Section 5 consists of the results and the discussion of each step in the proposed method. Conclusions are drawn in Section 6.

## 2. Related Work

Current methods for flooded road extraction can be divided into three categories by different data sources: rainfall simulation based on the rainfall time-series and assistance data such as DEM and drainage system data, change detection based on remote sensing images, and big data analysis based on geospatial big data with semantic information.

### 2.1. Extracting Flooded Roads by Rainfall Simulation

Rainfall simulation was initially used to predict the daily rainfall distribution by using atmospheric circulation in a large area over a long period of time [17]. Traditional rainfall simulation methods discovered the hidden relations between environmental indicators and the actual rainfall distribution to construct a prediction model [18,19]. However, the results of these methods often have large spatial scales and daily temporal resolution, which is difficult to apply to highly detailed urban areas.

Compared with large-scale forecasts of rainfall distribution, more conditions need to be considered for simulations with a high time resolution in small areas such as cities [20]. In addition to the need

for a rainfall time series, in order to improve the accuracy of simulation results, it is also necessary to better consider the impact of urban topography and urban rainwater drainage systems [21,22]. Chen et al. extracted 20 flooded buildings in University of Memphis Main Campus successfully by using rainfall data, DEM, and drainage systems. Similarly, this method is also suitable for the extraction of flooded roads in small areas [23]. Lee et al. proposed a method to simulate an urban flooded area by considering the road network and the building configurations in a larger area [24]. The improvement of this method involves replacing DEM with DSM to consider the height of the buildings in the study area, and in constructing a complex drainage model to approximate the reality.

There is no doubt that rainfall simulation is an effective method for detecting flooded areas and calculating the flooding depth and time. However, although all these rainfall simulations try to consider all the environment conditions in the model, the results are absolutely ideal because all the geographical processes are labile. In fact, the capacity of the drainage system varies from place to place, and even under high drainage pressure, loss can occur.

## 2.2. Extracting Flooded Roads by Remote Sensing Images

Traditionally, remote sensing images are used to classify the ground objects [25]. Unlike other objects, it is difficult to extract roads from remote sensing images at a high accuracy because the spectrum characteristics of urban roads are similar to other urban objects such as buildings [26]. Many researchers have focused on using high-resolution remote sensing images to detect urban roads in the space domain [27] and the frequency domain [28]. The extraction methods such as support vector machine (SVM) [29] and machine learning [30,31] are used in this field to improve the extraction accuracy.

However, because clouds can cover much of the ground, optical remote sensing images are difficult to obtain on rainy days, when light aircrafts can be dangerous to work on. To overcome the barriers to obtaining remote sensing images on rainy days, SAR (synthetic aperture radar) has been used to acquire radar remote sensing images in recent studies because of its strong penetrability and high resolution [32]. Tupin et al. [33] proposed a two-step method to extract the road networks by the linear features of SAR remote sensing images in 1998 and successfully applied it to several cases, which proved that SAR images could be used to extract roads. Wan et al. [34] extracted four flooded areas corresponding to flooded events that occurred in Guilin, China, by using SAR images and achieved a precision of over 0.75 and a recovery rate of about 0.99. Mason et al. [35] took advantage of high-resolution SAR images to detect flooded urban areas by using double scattering, which has a 100% classification accuracy of flooded double scattering curves.

SAR images can be used to extract roads and flooded areas, but the spectrum of slightly flooded roads is a complex mixture of an origin road spectrum and a water spectrum. Another way to extract flooded roads is by change detection between two SAR images of a sunny day and a rainy day. The problem of extraction by SAR images is that the flooding degree and the time at which roads are flooded cannot be measured.

## 2.3. Extracting Flooded Roads by Big Data with Semantic Information

In recent years, big data analysis has been a hot point to discover the hidden information amid huge amounts of data [36,37] and has been widely applied in the financial sector [38], to smart cities [39], in public management [40], and so on. Some researchers also try to mine the information of flooded roads through geospatial big data, such as SNS (social network service) data because geospatial SNS data contain spatial information and semantic information that describe the location and events of users. Song and Kim [41] proposed a big data analysis system for searching the flooded areas through weather data, road-link data, and SNS data uploaded by users in real time. This method constitutes most of the geospatial data, but the SNS data are highly subjective and unstable, so the results of the extraction can only be used for reference and are not able to supply the real flooding degree and time information from the semantic information.

## 3. Study Area and Data Source

Figure 1a shows the study area of Wuchang district, which is one of the three main urban areas in Wuhan, China. This area has a complex road network system and is prone to the formation of ponding in low-lying sections when extreme rainfall occurs. The study area is a typical area that covers 418 km$^2$, with an approximate length of 19 km and width of 22 km, and consists of different grades of urban roads.

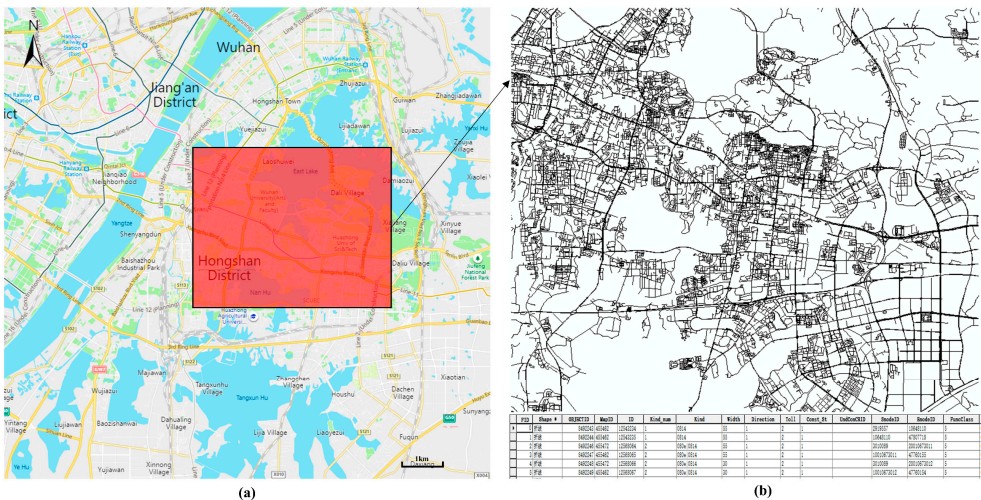

**Figure 1.** Study area and the vector road data. (**a**) The location of Wuchang district in Wuhan. (**b**) The vector road data with sematic information in Wuchang district.

In the study area, GPS trajectory data and vector road data were selected to analyze the flooded roads that cause traffic to be much slower than usual. Vector road data were downloaded from "Gaode Open Map Platform." Figure 1b shows the vector data, which consist of 34,601 roads with high spatial coordinate accuracy and basic semantic information such as ID, type, name, and so on. The data include all main urban roads and most other low-grade urban roads and are stored using the ESRI shapefile.

Taxi trajectories consist of GPS points that are uploaded from taxis in service in the study area. Each GPS point records the taxi ID, speed, direction, status, longitude, latitude, and upload time in sequence (Figure 2a). To distinguish the traffic status between sunny days and rainy days, we successively collected trajectories of 10,300 taxis in two days (48 h). Three million GPS points were collected on the sunny day (2016/7/5) and over 1 million were collected on the rainy day (2016/7/6), with a collection interval of 40–100 s.

The data used in this paper have the same coordinate system. Because of a special stipulation in China, the GCJ-02 geographic coordinate system is the primary coordinate system of taxi GPS trajectory points, which is different from the vector road data with the WGS-84 geographic coordinate system (Figure 2b) [42]. Since the position drift between the two coordinate systems is about 600 m, the taxi GPS trajectory points had to be projected to WGS-84. Finally, all data were projected onto the projection coordinate system, which entailed that the angle and distance of space objects were easy to calculate and that precision was high. Considering the applicability of projection coordinates, WGS84_World_Mercator was selected as the standard coordinate system of the experimental data (Figure 2c).

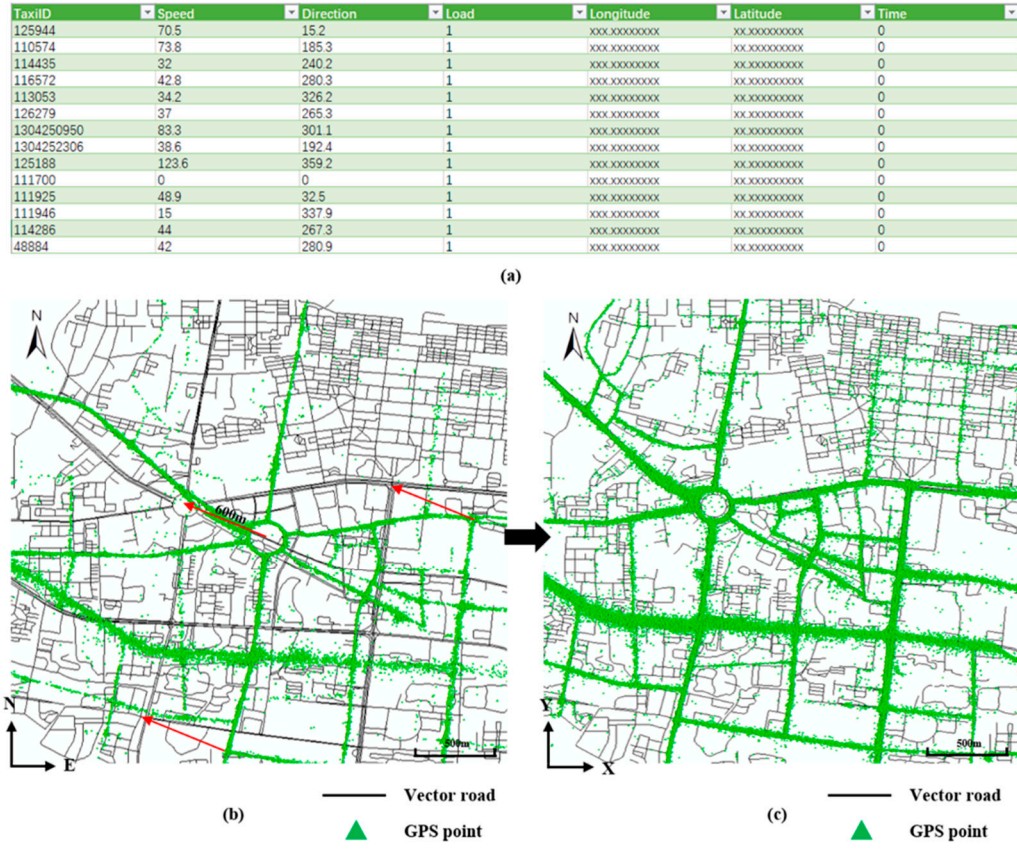

**Figure 2.** GPS point file and spatial coordinate transformation. (**a**) An example of a GPS trajectory point data file. (**b**) The location drift between GCJ-02 and WGS-84 is about 600 m. (**c**) GPS trajectory points and vector road data have the same location in the WGS84-World-Mercator projection coordinate system.

## 4. Methodology

The workflow of the proposed method (see Figure 3) is shown in the following steps, which aim to extract flooded roads in large urban areas.

Step 1: Calculate the probable inundated area in a grid view by comparing the differences between GPS trajectory points of normal and rainy days (Section 4.1).

Step 2: Calculate the probable flooded roads by using vector road data based on the method of map matching (Section 4.2).

Step 3: Measure the degree and time of flooded roads by using the probable inundated area and the probable flooded roads (Section 4.3).

The vector layer and grid layer each have their own advantages and disadvantages. In the grid layer, all GPS trajectory points have turned into a density grid to reflect the traffic flow of each area (grid), and the flooded area can be easily extracted based on the density grid. However, the extraction result of the grid layer is discrete and cannot be used to obtain precise road coordinates. On the contrary, the vector layer contains the semantic information and precise road coordinates of the urban roads. The fusion of the two different layers can take advantage of both data layers and generate a more accurate and complete result.

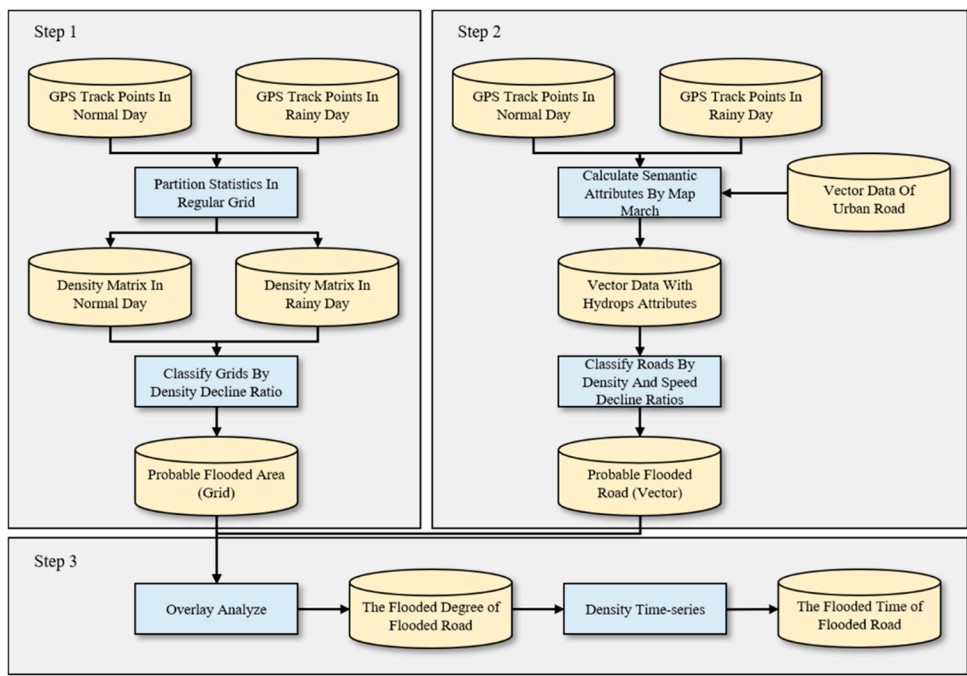

**Figure 3.** Workflow of flooded road extraction.

### 4.1. Extracting the Probable Inundated Area Based on a Grid

Millions of GPS trajectory points were selected to extract the flooded roads, but it is important to note that no single point reflects the status of the road. To discover the hidden value of trajectory points, the whole study area was divided into regular grids which is called point density grid, and the number of trajectory points in each grid is calculated [43]. Two density grids were calculated based on two datasets in order to discover the point density differences between normal and rainy days. The procedure is as follows:

#### 4.1.1. Confirm the Range and Scale of the Density Grid

The density grid range is the max area of one extraction, which is determined by the MBR (minimum bounding box) [44] of the vector road data. This information can be found in the file header of ESRI shapefile or can be calculated by the projection coordinate of each point in the road data file. In this paper, the range of the x-axis, $X_{min}$, $X_{max}$, and the range of the y-axis, $Y_{min}$, $Y_{max}$, are determined by the data source automatically, but it could also be set by the users with special needs.

Scale is the key parameter of this method: the larger the grid scale is, the more precise the result is. However, the grid scale is limited by the numbers of the GPS trajectory points in the study area, because when the number of points is insufficient, there will be a large number of empty grids in the road area. On the contrary, when the grid scale is at a low level, the density difference in each grid cell will be significantly reduced, which would probably lead to a negative result. Therefore, the parameter scale must fit the number of trajectory points, preferably no more than 10 m. After $\{X_{min}, X_{max}, Y_{min}, Y_{max}, \text{Scale}\}$ are obtained, the following formulas are used to calculate the rows and columns of the density grids.

$$\text{Rows} = \frac{Y_{max} - Y_{min}}{\text{Scale}} + 1; \text{Cols} = \frac{X_{max} - X_{min}}{\text{Scale}} + 1 \tag{1}$$

#### 4.1.2. Partition Statistics in the Grids

Define $P_N$ as the GPS trajectory point dataset on the sunny day and $P_R$ as the GPS trajectory points on the rainy day. Initialize two empty density grids $M_N, M_R$ with {Rows, Cols} calculated in

Section 4.1, in which the value of all grid cells is 0. The process of calculating $M_N$ is displayed as an example here. For each point $P^i$ in $P_N$, use Equation (2) to calculate the row index $m$ and col index $n$ of $P^i$ in $M_N$

$$m = floor\left(\frac{P^i - Y_{min}}{Scale}\right) + 1; \; n = floor\left(\frac{P^i - X_{min}}{Scale}\right) + 1 \quad (2)$$

The value of point density in the density grid located at $(m, n)$ increases by 1 (Equation (3)). The points located outside the density grid area are ignored in this procedure. M_R is calculated in the same way as $M_N$ and based on the dataset $P_R$ (Figure 4c):

$$M_N^{mn} = M_N^{mn} + 1 \quad (3)$$

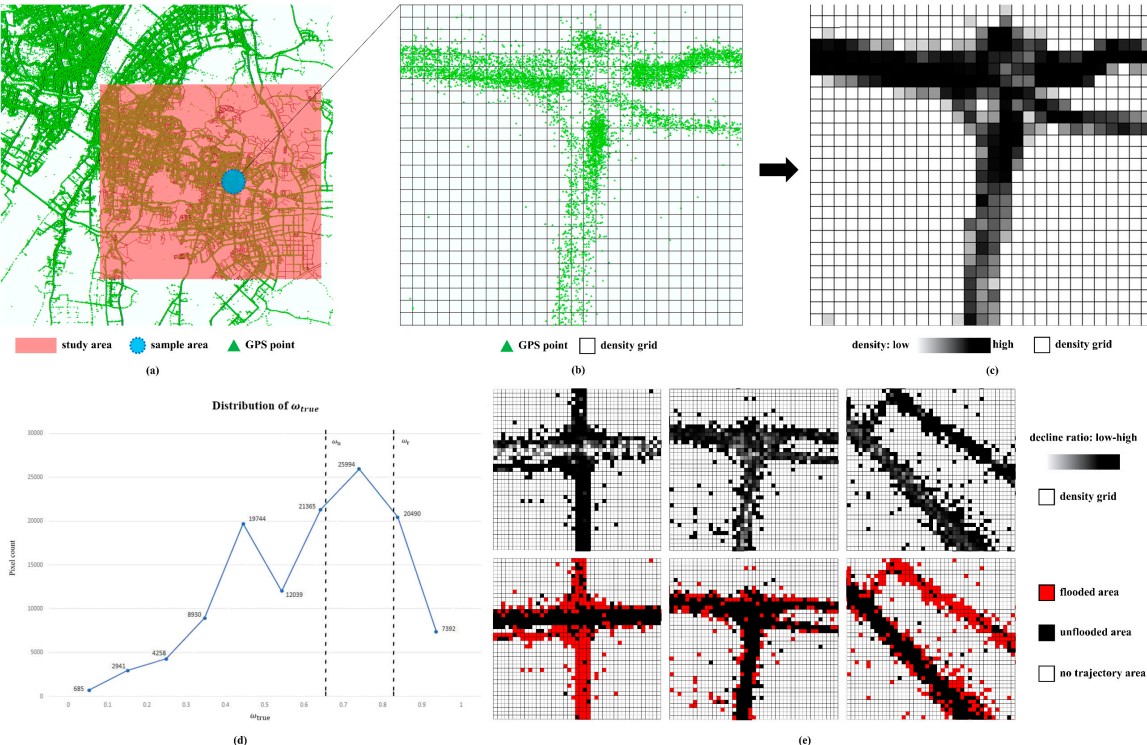

**Figure 4.** The process of extracting probable flooded areas. (**a**) The grid range based on vector road data and the location of example areas. (**b**) The GPS point distribution in example areas. (**c**) The density grid of example areas. (**d**) The distribution of the decline ratio $\omega_{true}$ and the location of parameter $\omega_n, \omega_r$. (**e**) Grid contains probable flooded area using $\omega_r$ in three different areas.

### 4.1.3. Extract the Flooded Area by the Density Grids

In this procedure, $M_N$ and $M_R$ are used to calculate the probable flooded area. There is no doubt that there is a conspicuous decline in the number of GPS points on the rainy day compared with the sunny day. This is because the willingness of people to travel on the rainy day decreased significantly, so a decline in density almost always occurred in each grid in the study area.

However, the flooded area can still be reflected by the density difference between $M_N$ and $M_R$ because the flooded area always shows a sharper decline than the normal area. Here, we can define two ratios $\omega_r, \omega_n$ to distinguish both density declines. First, if all the roads in the study area are not flooded, an assumption can be made that the density of the grids declined by the same ratio $\omega_n$, which is determined by the ratio of the number of points on normal and rainy days (Equation (4)):

$$\omega_n = 1 - count(P_R)/count(P_N) \quad (4)$$

Because no GPS points are located in the flooded area, it is easy to infer that the true decline ratio $\omega_{true}$ of the flooded grid is higher than $\omega_n$. On the contrary, the unflooded roads undertake the traffic pressure of the flooded road, so their $\omega_{true}$ is lower than $\omega_n$. The $\omega_{true}$ of each grid is calculated using Equation (5):

$$\omega_{true}^{mn} = 1 - M_R^{mn}/M_N^{mn} \tag{5}$$

Values greater than 1 or equal to 0 are ignored in the $\omega_{true}$ grid for a more precise $\omega_r$. Here $\omega_r$ is calculated by using Equation (6), where $E(\omega_{true})$ is the expectation of all $\omega_{true}$, and $\alpha$ is the extraction sensitivity range $[0, 1]$. If $\omega_r$ is greater than 1, $\omega_r$ is set to be equal to 1 (Figure 4d).

$$\omega_r = \frac{\sum \omega_{true}^{mn}}{n} + \alpha * \sqrt{\frac{\sum \left(\omega_{true}^{mn} - E(\omega_{true})\right)^2}{n}} \tag{6}$$

After obtaining $\omega_r$, the density grid can be divided into three kinds of area by comparing the density value between $M_N$ and $M_R$, and Equation (7) can be used to calculate the probable flooded areas $M_P$, where 2 means the flooded road, 1 means the unflooded roads, and 0 means other areas. Figure 4e shows the result of three different areas, where a red pixel represents an inundated area and a black pixel represents a common area.

$$M_P^{mn} = \begin{cases} 2 & M_N^{mn} * (1 - \omega_r) - M_R^{mn} > 0 \\ 0 & M_N^{mn} = 0 \cap M_R^{mn} = 0 \\ 1 & Other \end{cases} \tag{7}$$

### 4.2. Extracting Probable Flooded Roads based on Vector Road Data

Possible flooded areas have been already extracted in Section 4.1. However, the flooded areas in $M_P$ are discrete because of the characteristic of the grid data structure, which might result in an extraction result that cannot be converted to continuous roads and that suffers from a loss of sematic information. Vector road data have accurate spatial information and complete sematic information. The fusion of vector data and grid data can make good use of their advantages to extract flooded roads. The procedure is as follows:

### 4.2.1. Segment Roads in the Vector Data and Create a Spatial Index

The vector road data is usually stored in polyline ESRI shapefiles, where one polyline contains several single lines and a single line contains only two points. Single lines are used as the basic calculation units to improve the extraction accuracy, so all the polylines in the data file should be segmented. Because the number of road objects might increase several times after segmentation, a spatial grid index is needed to improve the efficiency of the algorithm [45]. The spatial index is created by using the same parameters as those used for creating the density grid, where each grid stores the indexes of single lines (Figure 5b).

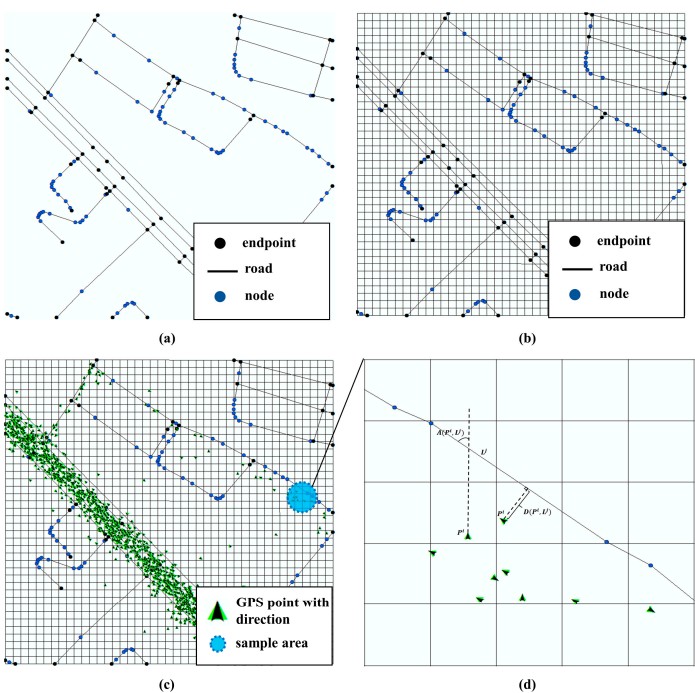

**Figure 5.** Overview of the process to create a spatial index and a map match. (**a**) Add endpoint at each node to split the polyline to a single line. (**b**) Construct a grid spatial index for the single line dataset to improve the efficiency of the process. (**c**) The distribution of GPS points with directions in the sample area. (**d**) The presentation of method $D(P, L), A(P, L)$ to calculate the distance and angle between the directed point and the single line.

### 4.2.2. Match GPS Trajectory Points to Single Lines by Map Matching

Map matching is useful for matching GPS trajectory points with their corresponding roads [46]. Each GPS trajectory point matches with its corresponding road, so the point density and average speed can be calculated on each line. The included angle and spatial distance between GPS trajectory point $P$ and single line $L$ are taken as the measurement of the matching degree. The spatial distance can be calculated using Equation (8), and the included angle using Equation (9):

$$D(P, L) = \frac{\left|\left(P_x - L_{sx}\right)\left(L_{sy} - L_{ey}\right) - \left(L_{sx} - L_{ex}\right)\left(P_y - L_{sy}\right)\right|}{\sqrt{\left(L_{sx} - L_{ex}\right)^2 + \left(L_{sy} - L_{ey}\right)^2}} \tag{8}$$

where $\left(P_x, P_y\right)$ are the $x$, $y$ coordinates of $P$; $(L_{sx}, L_{sy})$ are the $x$, $y$ coordinates of the start point in a single line; $(L_{ex}, L_{ey})$ are the $x$, $y$ coordinates of the end point in a single line.

$$A(P, L) = \left| P_{dir} - \text{atan}\left(\frac{L_{ey} - L_{sy}}{L_{ex} - L_{sx}}\right) + 90 \right| \tag{9}$$

where $P_{dir}$ is the instantaneous direction angle stored in the GPS data file; $\left(L_{sx}, L_{sy}\right)$ and $(L_{ex}, L_{ey})$ have the same meaning as the parameters in Equation (8); the range of $\text{atan}\left(\frac{L_{ey} - L_{sy}}{L_{ex} - L_{sx}}\right)$ is from −90 to 90. The range of the included angle is determined by whether the vector road data indicate a direction. If the vector road data indicate a direction, the range is 0–180; otherwise, the range is 0–90.

Calculate the position in the grid spatial index of each GPS trajectory point $P^i$ and find the single line set $L$ in its eight adjacent grids. For each single line $L^j$ in L, use Equation (10) to calculate the

matching degree G between $P^i$ and $L^j$. The higher the value of G, the better the matching between the point and the single line. The single line $L^j$ corresponding to the maximum G best matches to $P^i$.

$$G\left(P^i, L^j\right) = \cos\left(\frac{A\left(P^i, L^j\right)}{2}\right) / D\left(P^i, L^j\right) \tag{10}$$

where $A\left(P^i, L^j\right)$ is the included angle between $P^i$ and $L^j$, $D\left(P^i, L^j\right)$ is the spatial distance between $P^i$ and $L^j$, as shown in Figure 5d.

After map matching, all GPS trajectory points are matched to their best corresponding single line, and the point density and average speed of each single line can be calculated through the number of points and the driving speed of the matching dataset. Since the sampling interval of taxi trajectories in this paper is about 40 s, map matching in this procedure is relatively simple. If the GPS trajectory has a higher time resolution, more complex methods can be used to obtain a better map matching result [47].

### 4.2.3. Extract the Probable Flooded Roads

Based on the same vector road data, the point density and average speed of the sunny day and the rainy day can be obtained by using $P_N$ and $P_R$ (Figure 6a). Because the decline ratios of the point density and the average speed are necessary to calculate the probable flooded roads, this process uses parameter $\omega_r$ in Section 4.1 as the density decline ratio, and uses 0.5 as the constant ratio of speed decline. The reason why the speed decline ratio cannot be calculated by a traditional statistical method is that the ratio does not correspond to rainfall. Conversely, the drivers drive even faster when fewer cars are there on the unflooded road. Regarding flooded roads, the average speed shows a downward trend when they are gradually submerged by water. When the road is completely flooded, the average speed is 0; as the water begins to recede, the average speed returns to normal. Based on these rules, this paper simulates a simple linear model of speed decrease and increase to obtain the decline ratio of the average speed (Figure 6b).

If the point density and average speed of a single line meets one of the following conditions, it can be classified as a probable flooded road.

$$\text{(a) } L_{nd}^i * \omega_r - L_{rd}^i > 0$$

$$\text{(b) } L_{ns}^i * 0.5 - L_{rs}^i > 0$$

where $L_{nd}^i$ and $L_{rd}^i$ are the point densities of a single line on the sunny day and the rainy day, respectively; $L_{ns}^i$ and $L_{rs}^i$ are the average speeds of the single line on the sunny day and the rainy day.

As shown in Figure 7d, the red line in the figure indicates an obvious decrease in density and speed, which is also shown in Figure 7c. On the contrary, the density decline ratio of parallel black lines shows a normal downward trend, and the average speed is even faster than usual because of the decrease of cars.

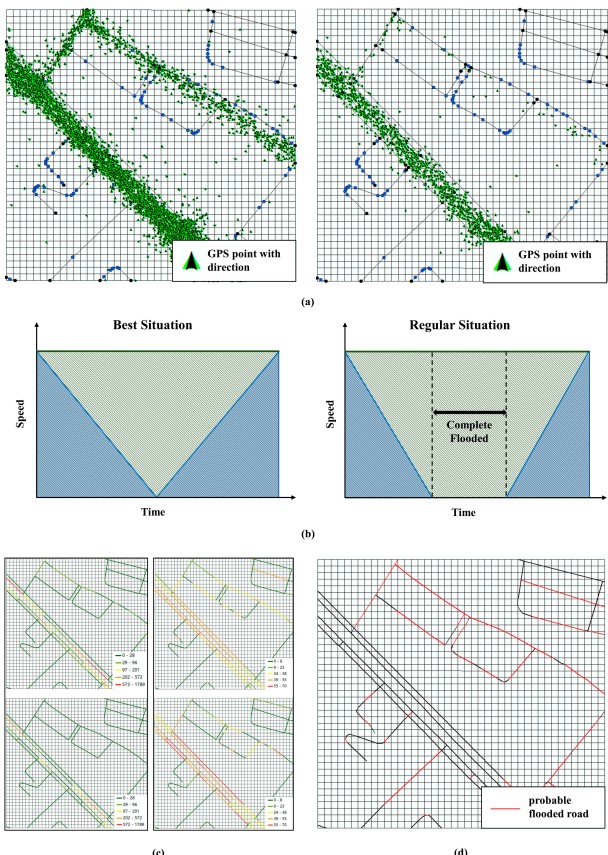

**Figure 6.** The workflow of extracting probable flooded roads by comparing the difference between normal and rainy days. (**a**)-Left shows the distribution of directed points on the sunny day; -right shows the distribution of directed points on the rainy day. (**b**)-Left is the best situation for flooded roads, and the average speed decline is about 0.5; -right is the regular situation, where decline is more than 0.5. (**c**)-Left shows the difference in density with the same symbol system on normal and rainy days; -right shows the difference in average speed with the same symbol system. In (**d**), red lines represent the probable flooded roads and black lines represent normal roads.

### 4.3. Extracting Flooding Degree and Time based on the Fusion of the Grid and Vector Road Data

Based on the all the procedures above, it is easy to determine that the extraction of flooded areas in Section 4.1 have no spatial continuity, and the extraction of flooded roads in Section 4.2 is not very reliable. By using the probable flooded roads and areas, the flooding degree and time can be calculated by data fusion to increase the extraction accuracy. In this paper, flooding degree is defined as the ratio of the actual flooded length to the total road length, so these two kinds of data provide possibilities. Grid data provide discrete flooding information while vector data provide precise spatial and sematic information. Equation (11) is used to calculate the flooding degree of each single line.

$$L_d = n_f / n_t \tag{11}$$

where $n_f$ means the inundated grid number, and $n_t$ means the total grid number of probable flooded roads through.

By calculating the flooding degree of each probable flooded road, this step reduces the misjudgment of the probable flooded roads caused by the weakness of map matching at low point density. It is not surprising that some roads are misclassified as probable flooded roads because there were few GPS trajectory points near them on the sunny day, making the density decline ratio less reliable.

The probable flooded roads that have a high flooding degree can be confirmed as the authentic flooded roads (Figure 7b).

The matching GPS points of each single line can be classified into different time positions in two continuous days to create the density time-series [48,49], and the flooding time analysis based on the road density time-series is very easy. As Figure 7c shows, the density–time series on the sunny day represents a conventional density fluctuation and remains zero for about 17 consecutive hours on the rainy day. This anomaly of continuous zeros indicates that the target road has been flooded during that period.

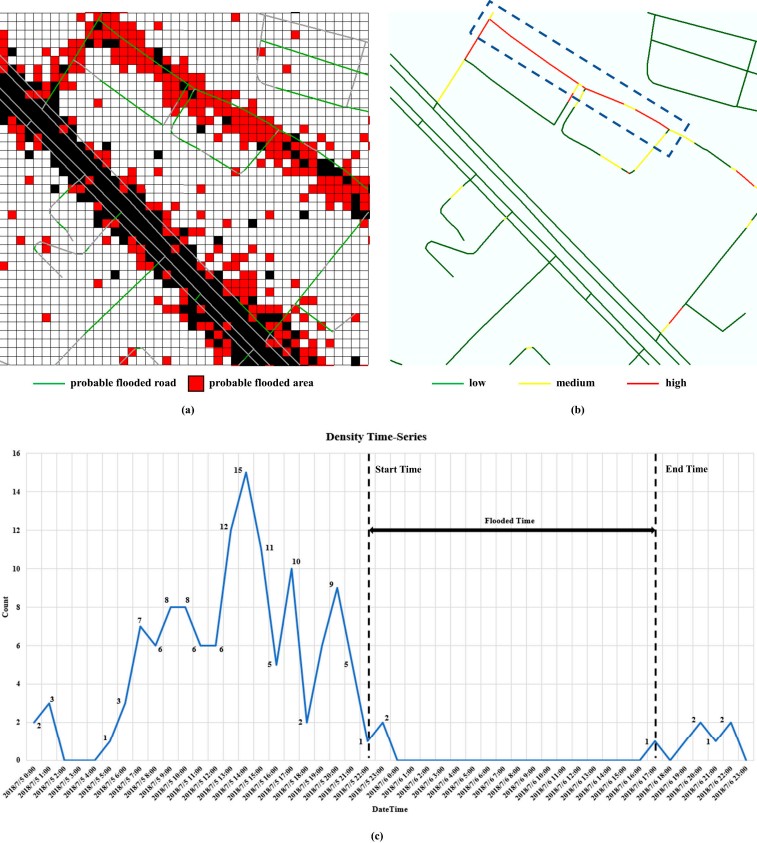

**Figure 7.** The process of extracting flooding degree and the time at which the road flooded. In (**a**), probable flooded roads overlay the probable flooded area to calculate the flooding degree by measuring the flooded percentage. (**b**) It shows the degree result of the process and divides degree into three levels: low, medium, and high. (**c**) It shows the flooding time of the flooded roads, which are selected in (b) by analyzing the density–time series with a 1-h interval.

## 5. Results and Discussion

### 5.1. Parameter Settings

The flooding degree and time extraction on urban roads in Section 4 requires certain parameters to ensure optimal performance in the procedures. It is hard for us to set constant parameters for all GPS trajectory point datasets, but the majority of point datasets have the same properties. Therefore, this paper proposes a default parameter group to satisfy common situations. All the parameters in the default parameter group can be modified to accommodate any other special situation.

The default parameter group consists of the scale for the density grid and the grid spatial index, a search range for map matching, and the time resolution of the time series. The scale is highly related to the number of GPS points and the accuracy of the user's requirement. Generally speaking, a larger scale corresponds to a higher accuracy but requires more GPS points. The sensitivity determines a

strict degree of extraction; a higher sensitivity leads to a smaller number of flooded roads, but the extraction result is more reliable. The search range for map matching is affected by the scale. A higher searching range corresponds to a higher accuracy of map matching but requires more calculating time. The time resolution of the time series is affected by the number of GPS points; a higher time resolution requires more GPS points to describe the density trend and leads to a better accuracy.

We designed several groups of parameters to explore the best default parameter group, which has a good balance between accuracy and efficiency. As Table 1 shows, the scale is set to $10 \times 10$ m, which is a suitable resolution to measure the density, the sensitivity is set to 0.33 to discover all the probable flooded areas [50,51], eight neighborhood pixels tolerate a 10–20 m location offset of GPS points, which is the situation for most points [52], and the time resolution is set to 1 h.

**Table 1.** Default parameters for extracting flooded roads.

| Parameter | Explanation | Default Value |
|---|---|---|
| Scale | Resolution of density grid and grid spatial index | Scale = 10 m × 10 m |
| Sensitivity | The sensitivity of flooded road extraction | $\alpha = 0.33$ |
| Range | Search range of map match | Range = 8 neighborhood pixels |
| Time Resolution | Time interval between two nearby record location in time-series | Resolution = 1 h |

*5.2. Experimental Results*

In this study, the GPS trajectory points on the sunny day and the rainy day were used to extract the flooding degree and time of each flooded road. First, the probable inundated areas were calculated by comparing the difference between the density grids of both days; second, the probable flooded roads were filtrated by map matching; finally, the flooding degree and time of each flooded road were extracted to verify the extraction result of the grid and vector. The total time consumption of this method was about 1116 s, the memory consumption was about 4 GB, and the data size was 100 thousand single lines and 4 million trajectory points.

Obtaining all real flooded roads on a specific day is challenging; however, we successfully collected some inundated areas and roads in Wuhan reported in the news on July 6, 2016 (http://www.sohu.com/a/101602173_347963), and tried to measure the extraction accuracy by comparing the extraction result with the real flooded roads. In Figure 8a, all 30 inundated areas and roads reported in the news are marked with transparent circles as the experimental control group. The experimental group also contains two pedestrian underpasses that were inaccessible to taxis, so Hongshan Square and Jiedaokou are marked with dark transparent circles and should be ignored in the following procedures.

In Figure 9, 8681 out of 110,828 single lines are classified as flooded roads by the proposed method, and all the circles in the left figure correspond to the reported flooded areas in Figure 8. Blue circles indicate successful extraction areas, and red circles indicate unsuccessful extraction areas. Compared with 28 available real flooded roads in the control group, 26 flooded roads were extracted successfully, and 2 were misclassified. The result shows that the recall accuracy of the proposed method in the study area is 92.85% without considering the omission error.

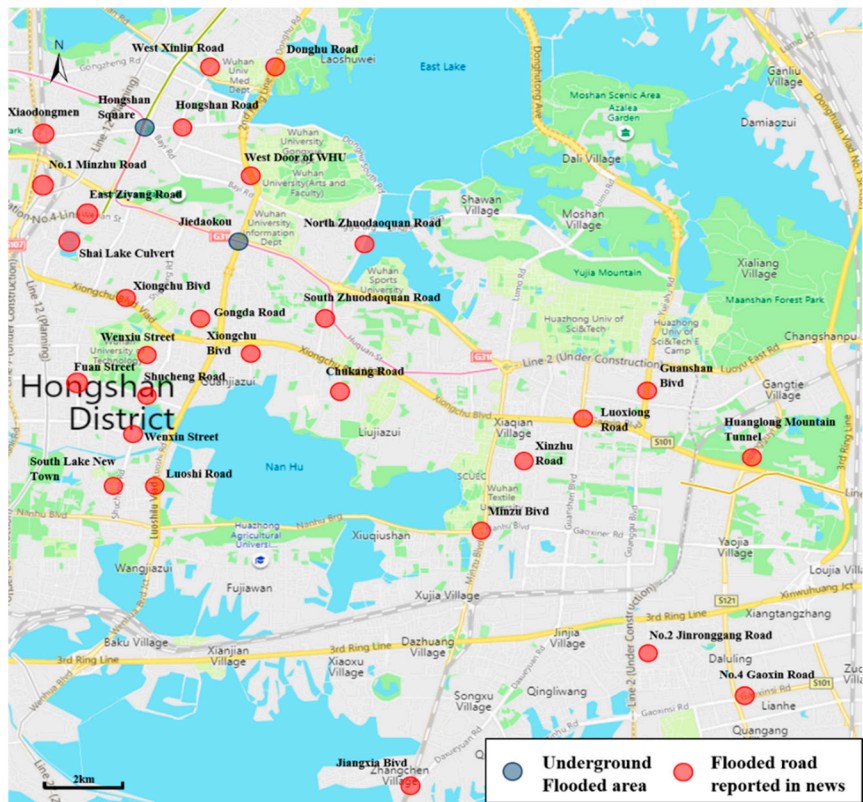

**Figure 8.** The 30 real flooded roads and areas as reported in the news.

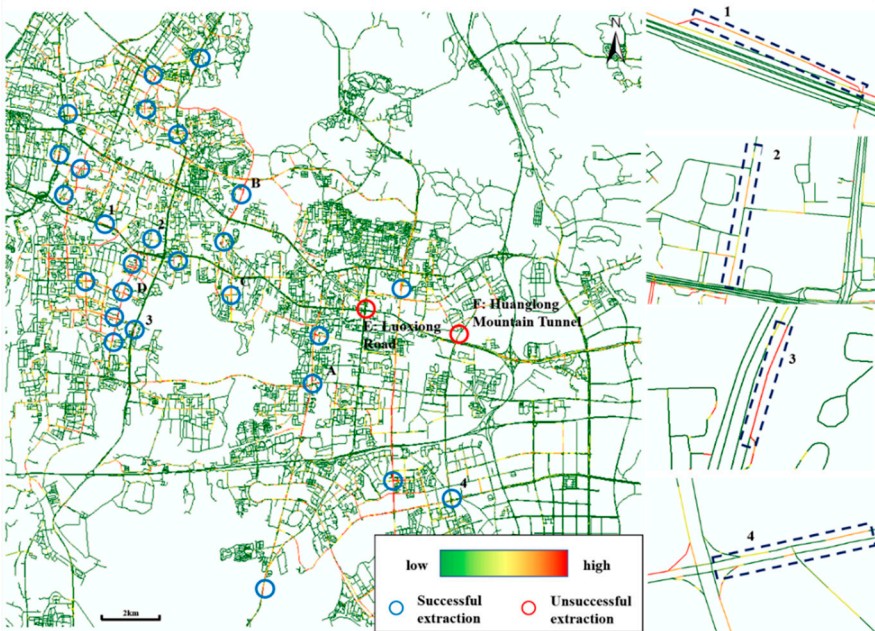

**Figure 9.** The result of the flooded road extraction method. Symbol A–F mark the six representative roads and 1–4 correspond to the detailed image on the right side.

As shown in Figure 10a–d, the density time-series of four flooded roads could easily directly reflect the flooding time. Obviously, all the density-time series have distinct continuous zeros or density at a very low level. Such a situation proves the correction of the extraction result. According to the density-time series, the flooding time range and the last flooding hours of the four roads are

displayed in Table 2. It could be derived that a higher flooding degree corresponds to a longer flooding time. The flooded roads with a high flooding degree mostly remained flooded for more than 10 h.

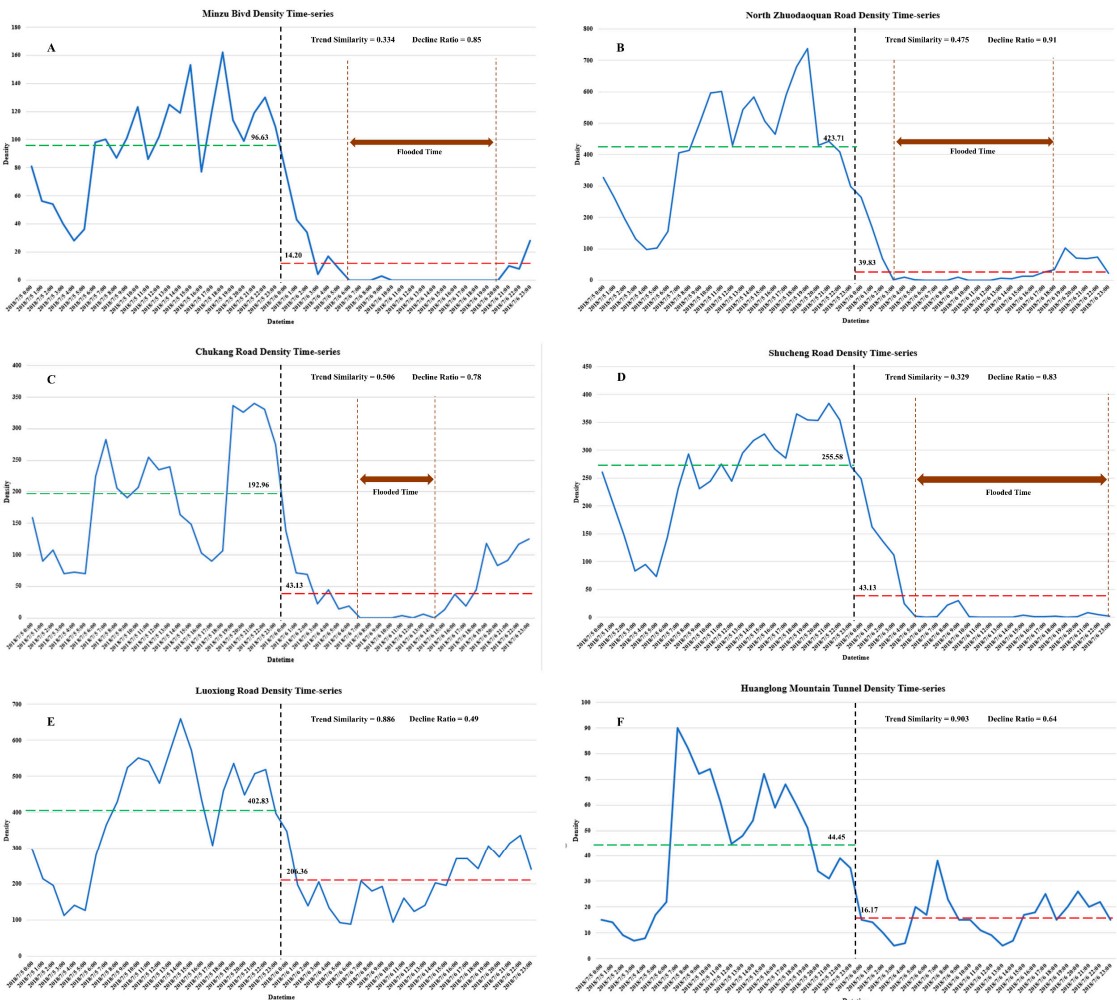

**Figure 10.** The time-series of representative roads. (**A**)–(**F**) are the time series of six representative roads that correspond to the symbols in Figure 9. The green lines and red lines indicate the average density of the sunny day and rainy day, respectively.

**Table 2.** Flooding degree and time of representative flooded roads.

| ID | Name | Degree | Time | Hours |
|----|------|--------|------|-------|
| A | Minzu Blvd | High | 6:00–20:00 | 14 |
| B | North Zhuodaoquan Road | High | 3:00–18:00 | 15 |
| C | Chukang Road | Medium | 7:00–14:00 | 7 |
| D | Shucheng Road | High | 5:00–23:00 | 18 |

*5.3. Discussion*

Although some flooded roads and areas have been reported in the news, it is still difficult to confirm that all the flooded roads are included. In addition, some roads are reported as flooded roads, but the ponds on the road did not affect the traffic. Such situations may cause two negative effects on the accuracy of the result:

1. Flooded roads were reported in the news but were not detected by the method. These roads often have a low flooding degree and are recognized as omission errors, but they do not actually damage the traffic.

2.　　Flooded roads were not reported in the news but were detected by the proposed method. These roads are considered as a misclassification in the common accuracy measuring method but they were actually flooded, and damaged the traffic.

In conclusion, due to the insufficient integrity of the control group, the simple accuracy above is lower than the actual value. In this section, we attempt to verify the accuracy of the extraction result.

### 5.3.1. The Flooded Roads Reported but Not Extracted

As shown in Figure 9, the two omitted classified roads are marked with red circles, indicating that these roads were reported in the news but not detected by the proposed method. The traffic flow is always in a regular daily pattern, so it was not difficult to find two situations occurring on the rainy day: The first is that the ponds on the road did not damage the traffic. The density–time series of this situation always showed a regular decline, but the trend did not substantially change. The second is one in which the traffic was damaged; in this case, the value in the density–time series sometimes declined sharply and the trend sometimes changed substantially. Thus, we considered the density decline ratio and the trend similarity to determine whether the ponds on the roads E and F have negative effects on the traffic flow.

The decline ratio could be calculated by the average density of the sunny day and the rainy day, and the trend similarity was measured by the cosine similarity [53] between the density vectors of both days. Cosine similarity ignores the absolute difference in values between two datasets and measures the difference by density trend changes.

As shown in Figure 10a–d, all four detected flooded roads have a sharp density decline that is much larger than the regular decline ratio $\omega_n$. In addition, the density trend similarity between the sunny day and the rainy day on these roads is very low since the heavy rainfall weakened the volatility of the regular characteristics at flooding time. The density–time series of roads E and F indicate that, although these two roads are reported as flooded roads in the news, they remained capable of moving traffic, which is reflected by the low density decline ratio and the high cosine similarity. Therefore, roads with good resistance stability, such as Luoxiong Road and Huanglong Mountain Tunnel, could not be classified as flooded roads on the rainy day, because they maintained a good carrying capability and patency.

### 5.3.2. The Additional Flooded Roads Extracted but Not Reported

The proposed method in this paper was also used to extract some flooded roads not reported in the news. In this section, we collect and analyze their density–time series to verify whether these roads were actually flooded or were misclassified by the proposed method. As shown in Figure 11, six extracted flooded roads in the study area are marked with black circles. Binhu Road and No. 3 Fozuling Road were selected as examples to prove that the additional extracted roads are indeed flooded.

The density–time series of Binhu Road is shown in Figure 11A. It is easy to find that the average density declined sharply on the rainy day and that the trend similarity was even lower than 0.1. This situation proves that Binhu Road indeed flooded from 8:00 to 23:00. Although the trend similarity of No. 3 Fozuling Road is at a medium level, the density reduction ratio of this road is even larger than 0.9, which means that the road was completely paralyzed on the rainy day. As shown in Figure 11B, considering that the density was very low from 1:00 to 5:00 on the sunny days, No. 3 Fozuling Road is confirmed as flooded from 6:00 to 23:00. Through the geographical location of the roads, Binhu Road is located north of Wuhan South Lake, so it is highly probable that Binhu Road will be subjected to flooding water on the rainy day. No. 3 Fozuling Road is located in a low-lying area, so it carries the precipitation pressure of the surrounding areas and seems easy to flood.

The other four roads marked with black circles in Figure 11 have similar characteristics. The roads in the extraction result not reported in the news could not be considered as misclassifications, because all these roads actually flooded at that time and had negative effects on the city traffic flow.

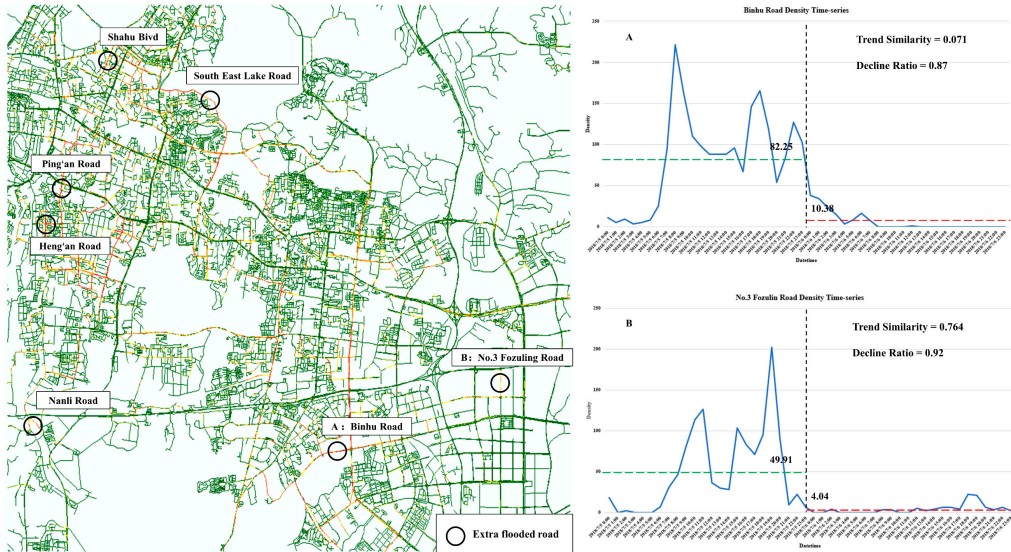

**Figure 11.** The additional flooded roads and their density–time series. (**A**) The density time-series of Binghu Road. (**B**) The density time-series of No.3 Fozulin Road.

### 5.3.3. Limitations of the Proposed Method

The proposed method can be used to automatically extract flooded roads at a high accuracy in a large study area. However, some limitations still remain. As the accuracy of the traditional simulation and remote sensing image-based method rely on the quality of the data, this method also depends on the number and positioning accuracy of GPS trajectory points. The more GPS trajectory points there are and the higher the positioning accuracy, the higher the extraction accuracy can become. On the contrary, if the number of points is not high enough, the statistical characteristic in the grid and vector will have a high degree of randomness and will not reflect the actual situation of the road. Decreasing the resolution of the density grid is one way of adapting to a low number of points, but this will cause a decrease in the resulting accuracy. If the GPS points have a high time resolution, interpolation can be used to solve this problem. However, the GPS trajectory point dataset is a kind of geospatial big data, and points are uploaded by urban taxis every day, so the quality of the data source seldom has a negative effect on the extraction result.

Another limitation is reflected by Figures 9 and 11. Most of the small roads are marked with green lines because the proposed method cannot detect roads that have no GPS trajectory points surrounding them. Such situations in the experiment results are due to taxis that could not drive into the residential quarter, resulting in no or very few GPS trajectory points. However, if the dataset is replaced with GPS trajectory points from private cars, the proposed method could still extract flooded roads in those communities.

All the limitations of the proposed method are due to the raw GPS trajectory data, so we recommend using GPS points with a high time resolution and a high positioning accuracy so that all the roads in the extraction area can be covered.

## 6. Conclusions

A new flooded road extraction method is proposed by taking advantage of the GPS trajectory point data. The proposed method can be used to successfully extract the flooding degree and the time at which roads were flooded in the study area, and the extraction result has a high similarity with respect to the flooded roads reported in the news. In addition, the omitted flooded roads in the extraction result prove, based on the density–time series, that the flooding on these roads do not actually have a negative effect on the traffic flow, so such roads could not be classified as flooded. Some additional flooded roads were extracted by the proposed method but were not reported in the news also proved

to be substantially flooded by analyzing the traffic flow on the rainy day. Compared with traditional flooded road extraction methods, the proposed method uses, as data sources, the GPS trajectory data, which can reflect actual road situations, and vector road data, which carries a high position accuracy with useful semantic information. In terms of extraction procedures, the proposed method has most of the advantages of grid layer extraction and vector layer extraction and calculates a more reliable result. The weakness of the proposed method depends on the quality of the GPS trajectory data and cannot extract flooded roads without GPS points around them. GPS trajectory points are uploaded every day automatically at no cost, and urban vector road data remain stable for a long period of time. If a city has enough taxi service and vector road data, the proposed method is easy to apply.

The GPS trajectory data is a type of big data that can be used to discover a great amount of hidden information, and such data have been used in several recent studies. The big geospatial data mining and data fusion have strong potential in terms of discovering unusual space–time events and of determining regular principles.

**Author Contributions:** Shiying She and Zhixiang Fang conceived and designed the experiment. Haoyu Zhong and Shiying She performed the experiments and analyzed the data. Meng Zheng and Yan Zhou contributed data and tools for analysis.

**Funding:** This work was supported in part by the National Key Research and Development Program of China (No. 2017YFB0503802), by the National Natural Science Foundation of China under Grant (41771473) and Grant (41231171), by the 2017 National Natural Science Foundation for Young Scholars of China (51708426), and by Special Funds for Basic Scientific Research Business in Central Colleges and Universities, Independent Scientific Research Project of Wuhan University in 2018 (2042018kf0250).

**Conflicts of Interest:** The authors declare no conflict of interest.

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
