# Peer review of "Extracting Flooded Roads by Fusing GPS Trajectories and Road Network"

_ijgi, doi:10.3390/ijgi8090407_

Round 1

Reviewer 1 Report

The presented content seems to be interesting and important in terms of classifying flooded areas. The described experiments are properly presented and include all necessary details. Nevertheless, the article merit - in many cases - is presented vaguely and contains lots of information, dependencies, relations or special cases correlated with each other, which makes the overall text perception difficult. Describing special regulations for using map projections in China would undoubtedly need some references. Extracting flooded areas by using GPS (maybe GNSS?) requires some explanations but not necessarily by giving many additional details. The general idea would need an in-depth explanation: what was the primary purpose of choosing such as methodology? The limitations provided in point 5.3.3 sound so serious that the main purpose of using the presented methodology would probably need a solid re-consideration. Tracking private cars - in terms of personal data protection - may not be possible at all. Some other roads can be dedicated only to pedestrians - how can they influence the overall result?

What is more, the method published in the journal should be characterised by a certain universality. How could the particular case considered in the paper be applied to other cities/regions of the World? From the technical point of view - the graphics need better resolution - some of them are difficult in perception. Also, colours and line thicknesses would need to be improved (ex. Fig. 9). To sum up, the article should be revised and presented more understandably.

Author Response

Response to Reviewer 1 Comments

Point 1: Nevertheless, the article merit - in many cases - is presented vaguely and contains lots of information, dependencies, relations or special cases correlated with each other, which makes the overall text perception difficult.

Response 1: Calculate in different layers and different data source might cause this problem. The whole proposed method can easily be understood as three steps:

GPS trajectory points to grid layer (density grid); GPS trajectory points and vector road data to vector layer (map match); The fusion of grid layer and vector layer to extracting flooded degree and time.

Point 2: Describing special regulations for using map projections in China would undoubtedly need some references.

Response 2: Because of the confidentiality policy of China, all the GPS trajectory points must be encrypted to create a position drift. This special encryption algorithm turns the real geographic coordinate system like wgs-84 to the fictitious coordinate system called gcj-02. Due to multi-source geospatial data could have the same coordinate system, the GPS trajectory points should be decrypted to origin coordinate system. An explanation from wikipedia has been cited in this paper to prove that the coordinate transformation is necessary (Reference No.52).

Point 3: Extracting flooded areas by using GPS (maybe GNSS?) requires some explanations but not necessarily by giving many additional details.

Response 3: Due to flooded areas always have the characteristics of lower traffic flow and driving speed. At the same time, the uploaded GPS trajectory points contain the sematic information such as speed, direction, so we consider that GPS trajectory points could reflect the real road situations actually. The explanation has been added at Row 64 in revised manuscript.

Point 4: The general idea would need an in-depth explanation: what was the primary purpose of choosing such as methodology?

Response 4: Vector layer and grid layer have their own advantages and disadvantages. In grid layer, all GPS trajectory points have turned into density grid to reflect the traffic flow of each area (grid) and flooded area could be easily extracted based on the density grid. However, the extraction result of grid layer is discrete and could not get the precise road coordinate. On the contrary, the vector layer contains the sematic information and precise road coordinate of the urban road. The fusion of two different layers could take advantage of both data layer and get a more accuracy and complete result. The explanation has been added at Section 4 Row 179.

Point 5: The limitations provided in point 5.3.3 sound so serious that the main purpose of using the presented methodology would probably need a solid re-consideration. Tracking private cars - in terms of personal data protection - may not be possible at all.

Response 5: There is no doubt that the accuracy of the proposed method relies on the quality of GPS trajectory points and urban vector road data. In Section 5.3.3, this paper considers that the more GPS trajectory points and higher positioning accuracy corresponding to a higher accuracy of extraction result. On the contrary, the less GPS trajectory points lead to a low accuracy extraction result, but it should be noted that the GPS trajectory point dataset is actually a kind of geospatial big data. Due to the data collection mode, the datasets always have a large amount of GPS trajectory points which satisfy the proposed method. About the question of user privacy, if researchers want to extract flooded roads in the community, they better use GPS trajectory points of private car than urban taxi. The private car data are always collected by navigation software and user information is encrypted to protect the personal privacy. So, analysis of private cars’ GPS trajectory points is no different with urban taxies’ points. Here, we put the use clauses of Gaode Map to prove the legitimacy of analysis (http://wap.amap.com/doc/serviceitem.html).

Point 6: Some other roads can be dedicated only to pedestrians - how can they influence the overall result?

Response 6: This might be a weakness of the proposed method. However, this method focuses on extracting flooded motorway for traffic management department. In all, this method tries to extract flooded roads which have bad effect on urban traffic but not sidewalk. About the influence on the overall result, due to sidewalk is not the target of the proposed method, I consider that the accuracy of the extraction result could not be affected by this problem.

Point 7: What is more, the method published in the journal should be characterized by a certain universality. How could the particular case considered in the paper be applied to other cities/regions of the World?

Response 7: The proposed method only needs two types of data: GPS trajectory points and vector road data. GPS trajectory points are uploaded by urban taxies and vector road data could downloaded from volunteer map platform like Open Street Map. So, if other cities have enough GPS equipped taxies and vector road data, the proposed method could apply to extracting flooded road. This question has been mentioned at Row 504 in this paper.

Point 8: From the technical point of view - the graphics need better resolution - some of them are difficult in perception. Also, colors and line thicknesses would need to be improved (ex. Fig. 9).

Response 8: Thank you for advising this problem, we have tried to improve the quality of the graphics in this paper. The detail images now have thicker line and more obvious color.

Reviewer 2 Report

The article is clear and balanced. I enjoyed reading it.

The method of extracting data is clearly described and understandable. The utilization is verified on test data from two days: sunny and rainy. I recommend in future another testing of the method on newer rainy days with collected GPS taxi data and field survey of real situation on roads.

Maybe typos:

row 345 5.1. parameters setting - capital P

row 358   number of GPS point(s)

row 499 serval factors - several factors?

Author Response

Response to Reviewer 2 Comments

Thank you very much for your recognition to this article, the question you raised has been modified in this article.

Reviewer 3 Report

The paper is an interesting one and worth a publication in IJGI, although some points need to be addressed.

The paper is an interesting one and tackle an issue that is of particular interest in research, that is the study of big data coming from GPS traces to infer derived information about users' behaviours or about the state of an infrastructure or a certain situation.  This is a particular relevant kind of research and its potential applications are enormeous.  Here the attention is on deriving from this kind of data, information about flooded roads in a region in China.  The application is an interesting one, but the emphasis should be given more on the importance of the big data analysis more than on the single application. In data explaning the data sources: over which period of time GPS data were traced from taxis? A further point to be better explained is related to the georeferencing and reference systems used. Why there is a 600 m lag between the different reference systems? Could it be better explained something related to the Chinese reference system and why there are mismatching between the other ones? Please also add scale bars and North arrows in the figures hosting a map. An overall comment is related to language. The English is very poor and required an important revision. A check and re-writing from a native English speaker would be appreciated.

Author Response

Response to Reviewer 3 Comments

Point 1: In data explaning the data sources: over which period of time GPS data were traced from taxis?

Response 1: The GPS trajectory points are uploaded every day since the GPS devices equipped and stored in the database. In this paper, we use GPS trajectory points from one sunny day and one rainy day to extract flooded road. So, the experiment in this paper use about 48-hour GPS trajectory points.

Point 2: A further point to be better explained is related to the georeferencing and reference systems used. Why there is a 600 m lag between the different reference systems? Could it be better explained something related to the Chinese reference system and why there are mismatching between the other ones?

Response 2: Because of the confidentiality policy of China, all the GPS trajectory points must be encrypted to create a position drift. This special encryption algorithm turns the real geographic coordinate system like wgs-84 to the fictitious coordinate system called gcj-02. Due to multi-source geospatial data could have the same coordinate system, the GPS trajectory points should be decrypted to origin coordinate system. About this problem, we cited a reference article in this paper to explain (Reference 52).

Point 3: Please also add scale bars and North arrows in the figures hosting a map.

Response 3: Thank you for this kind advise, the scale bars and north arrows have been added to the maps (Figure 8 9 11) in this paper.

Round 2

Reviewer 1 Report

The reviewer wishes to thank the authors for their informative responses. In the same time, it can be spotted, that the authors introduced all suggested explanations as well as they completed desirable information. Regarding that, the reviewer has neither additional questions nor objectives. After checking the English language of the text (especially spelling of the words - ex. "rader" instead of "radar" etc.) it can publish in the present form.

Author Response

Point 1: After checking the English language of the text (especially spelling of the words - ex. "rader" instead of "radar" etc.) it can publish in the present form.

Answer 1: Thank you for your kind comments on this paper. We have examined the spelling of this paper and the language of this paper has been modified by professional English language editor.

This manuscript is a resubmission of an earlier submission. The following is a list of the peer review reports and author responses from that submission.